# Surface Hygiene Evaluation Method in Food Trucks as an Important Factor in the Assessment of Microbiological Risks in Mobile Gastronomy

**DOI:** 10.3390/foods12040772

**Published:** 2023-02-10

**Authors:** Michał Wiatrowski, Elżbieta Rosiak, Ewa Czarniecka-Skubina

**Affiliations:** Department of Food Gastronomy and Food Hygiene, Institute of Human Nutrition Sciences, Warsaw University of Life Sciences (WULS), 166 Nowoursynowska Str., 02-787 Warsaw, Poland

**Keywords:** food trucks, hygiene, Petrifilm^TM^, reference method, ATP, TVC, pathogenic microorganisms

## Abstract

Street food outlets are characterised by poor microbiological quality of the food and poor hygiene practices that pose a risk to consumer health. The aim of the study was to evaluate the hygiene of surfaces in food trucks (FT) using the reference method together with alternatives such as Petrifilm^TM^ and the bioluminescence method. TVC, *S. aureus*, *Enterobacteriaceae*, *E. coli*, *L. monocytogenes*, and *Salmonella* spp. were assessed. The material for the study consisted of swabs and prints taken from five surfaces (refrigeration, knife, cutting board, serving board, and working board) in 20 food trucks in Poland. In 13 food trucks, the visual assessment of hygiene was very good or good, but in 6 FTs, TVC was found to exceed log 3 CFU/100 cm^2^ on various surfaces. The assessment of surface hygiene using various methods in the food trucks did not demonstrate the substitutability of culture methods. Petrifilm^TM^ tests were shown to be a convenient and reliable tool for the monitoring of mobile catering hygiene. No correlation was found between the subjective visual method and the measurement of adenosine 5-triphosphate. In order to reduce the risk of food infections caused by bacteria in food trucks, it is important to introduce detailed requirements for the hygiene practices used in food trucks, including techniques for monitoring the cleanliness of surfaces coming into contact with food, in particular cutting boards and work surfaces. Efforts should be focused on introducing mandatory, certified training for food truck personnel in the field of microbiological hazards, appropriate methods of hygienisation, and hygiene monitoring.

## 1. Introduction

Street food refers to ready-to-eat food products, including fruits and vegetables, that are sold in public places, mainly on the streets. Street vending is common in both developing and developed countries. The largest number of such outlets is found in Africa, Asia, and Latin America. It often belongs to the informal food supply sector, which is characterised by unregulated production and hygiene practices [1]. In the last few decades, street food has also become popular in Europe [2,3].

Previous studies on street food establishments focus primarily on consumer choices and frequency of use of this form of facilities [2,3,4,5,6,7,8], determining the nutritional value of the meal, the risk of developing diet-related diseases [9,10,11], assessing hygiene practices, and the risk of health hazards to consumers [7,12,13,14,15,16,17,18,19,20,21,22,23,24,25], which have become important aspects of public health [26,27].

The majority of studies have used visual appraisal to assess the hygiene status of street food outlets. In previous studies [2,24,28,29,30,31,32,33], it has been stated that vendors are aware of food hygiene and respect good hygiene practices, but street food vendors with elementary education levels should undergo basic training on food hygiene. Thus, the need for health education to improve vendors’ knowledge on hygiene practices and food safety becomes apparent [34].

In many parts of the world, street food has been linked to diseases that pose a threat to public health [35,36]. Many authors [17,19,27,28,29,30,31,32,33,34,35,36,37,38,39,40,41,42,43,44,45] indicate that, due to the low microbiological quality of street food meals, their consumption may pose a risk of foodborne disease.

Pathogenic microorganisms such as *Escherichia coli*, *Bacillus cereus*, *Clostridium perfringens*, *Staphylococcus aureus*, and *Salmonella* spp. are usually present in street food [7,46]. Among the street-vended products that have been found to be contaminated with aerobic microflora, *S. aureus*, *Salmonella typhi*, or *E. coli* and the coli group of bacteria are samples of tomato sauce, rice balls, and peanut soups [47] and barbecue chicken [48,49,50,51,52,53]. *E. coli*, *S. aureus*, *Pseudomonas aeruginosa*, *Klebsiella*, and coliform bacteria have also been found to contaminate ready-to-eat products such as sandwiches, panipuri, momos, chola, samosa, vegetable salads, packaged fried rice, and egg burgers [54,55,56,57], as well as chicken meat [58]. The results also indicate poor microbiological quality (TVC, *E. coli*) of meat-based ready-to-eat fast food items (chicken sandwiches, chicken burgers, and hot dogs) sold on the streets [59] and in grilled chicken [60]. Fungi such as *Aspergillus flavus*, *A. niger*, *A. candidus*, *Cladosporium herbarum*, *Necrospora crassa*, *Penicillium citrinum*, *Fusarium*, *Mucor*, and *Rhizopus* have been identified in various forms of street food [61]. In 118 cooked and uncooked falafels, there was a higher than recommended microbial TVC (10^8^ CFU/g), coliform count (3.7 × 10^3^ CFU/g), and mould count (1.3 × 10^3^ CFU/g). Such contamination results in a significant risk of intake of these products [62]. TVC greater than 5 log CFU/g and coliform counts greater than 4 log CFU were considered unsatisfactory and are indicative of poor hygiene standards [60]. The presence of coliforms, including faecal coliforms or *E. coli,* indicates the adoption of poor hygiene practices or unhygienic conditions during food processing [63,64]. The authors have identified the reasons for this to be the limited education of vendors, a lack of training in good hygiene practices, and inadequate food preparation temperatures, all of which result in poor microbiological quality of the final product. Outbreaks of microbiological contamination of street food in many countries have also been linked to poor water quality [65] and poor quality of food ingredients [13,66,67,68], as well as less attention being paid to hygiene in street food outlets.

Several authors [69,70,71,72,73,74] identify kitchen utensils and cutlery as a source of serious microbiological risks: spoons, knives, cutting boards, and plates, as well as the hands of employees. These surfaces were found to be contaminated with high numbers of microorganisms, e.g., *B. cereus*, *E. coli*, *Shigella sonnei*, *Clostridium perfringens*, *Salmonella* spp., and *S. aureus*. Most of the results of microbiological studies of street food outlets have been carried out in developing countries, whereas only a small number of these studies have focused on the assessment of these types of establishments in Europe.

Ensuring appropriate production quality and health safety in mobile catering facilities is determined by the specific nature of the work in this type of establishment. Above all, this type of establishment is characterised by difficult working conditions, often without access to running water; a lack of regularity in the work; a variety of foods offered; a rush to prepare meals and carry out hygiene procedures; often unsatisfactory equipment in the production and serving area; and little attention paid to ensuring food quality and safety. Additionally, the small number of staff recruited, the need for multitasking, the unavailability of employees with adequate professional training in catering technology and food hygiene, the high staff rotation resulting in the constant need to train new employees, the low social status of the employees, and their low salaries all add to the challenges faced [74].

The aim of this study was to assess the status of surface hygiene in mobile food establishments using different analytical methods, traditional and rapid diagnostic methods, and related sampling techniques. The paper addresses the issue of whether and which alternative methods of hygiene assessment might be useful if applied in food trucks, which surfaces in food trucks are best inspected using the available hygiene assessment methods, and with what frequency these surfaces should be inspected with these methods. Our study fills a research gap in the microbiological quality of European food trucks, as well as in comparing and assessing the effectiveness and suitability of traditional and alternative methods of ensuring the health safety of food truck consumers.

## 2. Materials and Methods

### 2.1. Materials

The material for the study consisted of swabs and samples taken from 100 different service areas in 20 mobile food catering establishments operating in Warsaw, Poland, in cases in which the owners agreed to provide samples. The food served in the surveyed facilities was prepared in an eat-in and take-out format (Appendix A). The material for testing in each establishment was taken from five surfaces: a shelf of a refrigerator, a cutting board, a small utensil such as a knife, a serving surface, and a worktop surface.

Samples were taken before the opening of the establishment and the start of production. According to the owner’s declaration, the surfaces were previously cleaned and sanitised. Immediately after collection, the test material was transported to the laboratory under refrigeration [75] and analysed microbiologically. In addition, a subjective visual assessment of the condition and cleanliness of the equipment and work surfaces was conducted by the same person who collected the surface samples.

### 2.2. Microbiological Methods

Three methods were used to assess microbiological contamination. Two direct culture methods were selected for the detection of microbiological contaminants: the reference swabbing method (plate method) and the alternative contact method (agar swabs) using Petrifilm (3M™Petrifilm™, Kajetany, Poland) plates and the indirect method of bioluminescence measurement of the microbial cell energy metabolite ATP (adenosine 5-triphosphate). A scheme of the tests conducted is shown in Figure 1.

#### 2.2.1. Reference Plate Method

The swabbing method, a modification of the classic stamp method, involves swabbing from a limited template (10 × 10 cm^2^) area with a sterile swab (PROBACT medical, Heywood, Lancashire, UK). The technique of swabbing was standardised with a zig-zag movement of the swab in 4 planes: vertical, horizontal, and two diagonal planes on the template [75]. The swab was transferred to a sterile extender and vortexed (3 × 5 s) (LP Vortex Mixer, Thermo Scientific, Waltham, MA, USA), and microbiological surface inoculation was performed. The limiting template was sterilised with a burner flame before each material collection. The characteristics of the analyses performed using the reference method are presented in Table 1.

#### 2.2.2. Alternative Contact Method—Petrifilm

Petrifilm tests (3M™Petrifilm™) are ready-to-use plates with dehydrated conditioner dedicated to the detection of specific microorganisms or groups of microorganisms. The Petrifilm plate consists of two parts. The lower part contains the culture medium, which, after rehydration (30 min) with a sterile water conditioner, is transferred to the upper film to make an imprint on the surface to be tested. After collection, the tests were incubated according to the manufacturer’s instructions. Analyses were carried out according to the characteristics shown in Table 2.

In order to unify the expression of microbial contamination of surfaces assessed by different methods, microbial contamination was expressed as log CFU/100 cm^2^. The obtained results were multiplied by the appropriate coefficient (2.5-EL, 3.3-STX lub 5-PAC, EB, EC). For the Petrifilm STX plates, when uncharacteristic growth was observed, detection of colonies belonging to *S. aureus* was carried out using a disc containing bule-O toluidine, an enzyme produced by *S. aureus*. Readings were taken in accordance with the 3M™Petrifilm™ Interpretation Tables [75].

#### 2.2.3. Interpretation of Culture Methods Results

Results obtained in colony forming units (CFU) per plate area were converted to log CFU/100 cm^2^. For the assessment of the surface hygiene with the reference method, the acceptable contamination on surfaces in contact with food in the case of TVC is log 3.0 CFU/100 cm^2^; for *S. aureus*, it is log 4.0 CFU/100 cm^2^; and any presence of *E. coli*, *Enterobacteriaceae*, *Salmonella* spp., and *L. monocytogenes* is unacceptable [82].

#### 2.2.4. ATP Bioluminescence Measurement

Indirect method tests were performed using a Clean-Trace™ NG (Noack Polen) luminometer Clean-Trace™ NG (3M Health Care, Neuss, Germany) and tests Clean–Trace Surface ATP (ULX100 3M). Swabs were collected from surfaces using a limited metal template (100 cm^2^) which was sterilised before each collection.

We use an indirect method to assess hygiene by ascertaining the level of the bioluminescence of adenosine 5-triphosphate present on the surface. ATP is a nucleotide which is the carrier of free energy in every living cell of the sampled biological material. The concentration of ATP in a swab sample is directly proportional to the level of light emitted. A high level of ATP may, therefore, be evidence of the presence of microorganisms or organic remains on the tested surface in a catering establishment.

#### 2.2.5. Interpretation of Bioluminescence Measurement Results

On the basis of the findings of Griffith et al. [83], an ATP value of up to 500 RLU was taken as a realistic limit for clean surfaces.

### 2.3. Statistical Methods

Statistica 13.3 PL (TIBCO Software Inc., 2017 WA 98109, Seattle, WA, USA) software was used to compare the results and perform cluster analysis. To determine the difference between the samples, one-way ANOVA analysis of variance was used. The significance of differences between individual means was determined using Tukey’s post-hoc test (RIR). An α value of 0.05 was used. The cluster analysis method were used to classify the results of microbiological analyses of surfaces [84]. The distance between clusters was measured by Euclidean distance function, whereas the Ward method was used to bind the clusters. The Ward method uses the assumptions of variance analysis and aims to minimise the sum of deviations within clusters. As a result of joining cluster pairs, the pair that gives the cluster with the minimum differentiation is chosen. ESS (Error Sum of Squares) is a measure of the difference from the mean value. The linear coefficient of Pearson to assess the correlation between the used analytical methods was used (previously, the Shapiro–Wilk test was performed). Correlation heatmap was plotted in SRPlot at https://www.bioinformatics.com.cn/en (accessed on 30 December 2022).

## 3. Results

### 3.1. Visual Assessment of the Surfaces in Street Food Outlets

The results of the visual assessment of surface cleanliness in street food outlets are presented in Table 3. Only three food trucks received a high visual assessment of the surface (5 points). In 10 FTs, not all surfaces were clean, and they were rated as having good hygiene (4 pts). In turn, surfaces in contact with food in six FTs were rated as having low satisfactory hygiene (3 pts) or unsatisfactory hygiene (2 pts). One of the assessed food trucks was rated as having a poor hygiene state (1 point).

### 3.2. Presence of Indicator Microorganisms on Surfaces in Street Food Outlets

#### 3.2.1. Total Viable Count

Appendix A and Figure 1 show the results of microbiological analyses of the presence of the total number of aerobic mesophilic microorganisms on the tested surfaces. The analyses were conducted using the following methods: imprinting with the use of Petrifilm tests and the reference traditional method. The maximum contamination limit was log 3 CFU/100 cm^2^ considering the possibility of pathogenic microorganisms and faecal contamination among the detected microorganisms.

Results obtained from the surfaces of refrigerators (rP, rR), cutting boards (cP, cR), knives (kP, kR), and serving (sP, sR) and working boards (wP, wR) from 20 food truck outlets varied significantly; *p* < 0.05 (Figure 2a,b; Appendix A).

The results of the analyses provided by the reference method were in the range of 0–3 log CFU/100 cm^2^. The exceptions were FT2, FT10, and FT15, in which log 5.22, 5.85, and 6.57 CFU/100 cm^2^ were detected on the serving board, knife, and working board, respectively.

Regarding the contamination evaluated using the Petrifilm method, TVC values above the acceptable level were found on the three tested surfaces. On the serving board surface, contamination of log 5.39–6.73 CFU/100 cm^2^ was found in FT3, FT4, and FT10, respectively. On the working board surface, contamination of log 5.27–6.93 CFU/100 cm^2^ was found in FT2–FT5 and FT10, whereas in FT19, the accepted level was slightly excessive (log 3.32 CFU/100 cm^2^). The cutting board surface in FT2, FT5–FT6, and FT10 was contaminated with a TVC of log 4.74–6.73 CFU/100 cm^2^.

#### 3.2.2. *Enterobacteriaceae* and *E. coli* Bacteria

*Enterobacteriaceae* and *E. coli* were detected via growth culture methods in 50% of the tested food truck outlets (Table 4). Using imprint plate methods, *Enterobacteriaceae* contamination was found at levels not exceeding log 1 CFU/100 cm^2^ for 15 surfaces in eight food trucks, and contamination was found at levels not exceeding log 2 CFU/100 cm^2^ on 5 surfaces in FT1 and FT2. In FT10, *Enterobacteriaceae* were found on all surfaces tested: refrigerator, cutting board, knife, serving board, and working board at levels of log 2.86, 2.90, 3.02, 2.92, and 6.99 CFU/100 cm^2^, respectively. By using a conventional swab and the reference analysis method, *Enterobacteriaceae* at levels not exceeding log 1 CFU/100 cm^2^ were detected on eight surfaces in FT1–FT5. In food truck outlet FT1, *Enterobacteriaceae* were found at a level not exceeding log 2 CFU/100 cm^2^. Contamination with bacteria belonging to the genus *Enterobacteriaceae* indicates the risk of pathogens such as *Salmonella* spp., *Shigella* spp., *Klebsiella* spp., *Cronobacter* spp., *Serratia* spp., and *Citrobacter* spp. Therefore, the hygiene standard for these microorganisms was adopted as “zero tolerance”. In food trucks F7, F8, and F13–F20, *Enterobacteriaceae* and *E. coli* were not detected; therefore, they are not included in Table 4.

The presence of *E. coli* was found by both methods using Petrifilm plates and the reference analysis method on five and two surfaces tested in FT1 and FT2, respectively. No *E. coli* was found in the remaining food truck outlets, that is, FT3–FT20.

### 3.3. Presence of Pathogenic Microorganisms on Surfaces in Street Food Outlets

*Salmonella* spp., *L. monocytogenes*, and *Staphylococcus aureus* were detected and quantified using the reference method. Meanwhile, environmental *Listeria* and *S. aureus* were detected using an alternative method. Petrifilm plates are not available for analysis for *Salmonella* spp. The reference method did not identify *Salmonella* spp. or *L. monocytogenes* in any of the food trucks evaluated. An alternative method using Petrifilm plates was used to detect environmental *Listeria* spp. Using this method, *Listeria* spp. were detected in Food Truck No. 5 (refrigeration and serving board), No. 6 (refrigerator), and No. 12 (working board) in numbers not exceeding log 2 CFU/100 cm^2^.

Figure 3 and Appendix A, presents the results of microbiological analyses of the prevalence of *S. aureus* on tested surfaces in the food trucks.

The presence of vegetative *S. aureus* cells does not represent an immediate threat from the pathogen because what matters is the number of colony-forming units on a given surface. The problem is a concentration above log 4 CFU/1 cm^2^ on a given surface or directly in the food (g/mL), at which point *S. aureus* produces an exogenous, thermostable toxin that is a threat to food safety and consumer health. The results of the culture-based analyses (reference and alternative) were statistically different (*p* < 0.05) (Appendix A). The median value of the results obtained, regardless of the method used, was below log 1 CFU/100 cm^2^. The scores in the third quantile as well as the outliers also did not exceed log 3 CFU/100 cm^2^. Based on the results obtained, it was observed that the analysed surfaces in the investigated food truck outlets varied in hygienic status, which is evidenced by a significant spread in quantiles 2 and 3 and by outliers.

### 3.4. Evaluation of Hygiene in Street Food Outlets Using the ATP Method

The indirect method of hygiene evaluation using the measurement of adenosine 5-triphosphate levels is one of the fastest measurement methods compared to culture methods, with results obtained in just a few minutes. Therefore, an experiment was carried out to correlate the results so obtained with those from culture methods. The metabolite that is analysed is generated from the decomposition of a high-energy compound contained in microbial and eukaryotic cells. The result obtained is directly proportional to the level of ATP content of microbial and organic sources. Figure 4 shows a characterisation of the level of ATP occurrence on the food truck outlet surfaces studied. On 25% of the examined surfaces, the RLU level exceeded the tolerance value by more than 500 RLU/100 cm^2^. With the exception of FT2, ATP was found on all of the tested surfaces in the range of 0–1000 RLU (Relative Light Unit) (Figure 4a,b). In the case of the exception of food truck outlet FT2, ATP levels in the range of 4000–7000 RLU were found on the refrigerator and on working and cutting board areas, which significantly affected the distribution of values of the results obtained (Figure 4b).

### 3.5. Evaluation of the Suitability of Methods to Measure the State of Hygiene in Street Food Outlets

In the cluster analysis that was conducted, the length of the bond directly represents the level of contamination evaluated on the surface. For TVC analyses (Figure 5, upper) using the Petrifilm method, the surfaces tested were grouped into two clusters. The first included the refrigerator and knife surfaces as surfaces on which the accepted level of log 3 CFU/100 cm^2^ was not exceeded, whereas the second included the other surfaces tested, with the cutting board being separated out as a separate cluster, indicating a different microbiological quality to that of the serving and working boards. In contrast, three clusters were identified for the reference method. The surfaces of the refrigerator and the cutting board were found to be a concentration cluster of surfaces for which none of the food truck surfaces was found to exceed the accepted level of contamination. A serving board surface with a contamination level of log 5.22 CFU/100 cm^2^ (FT3) was also included in this cluster. Clusters II and III are surfaces on which TVC contamination was found at log 5.87 and 6.57 CFU/100 cm^2^, respectively.

The cluster analysis that was conducted allows quick identification of clean and contaminated surfaces. Furthermore, it reveals differences in surface contamination depending on the analysis method used. In the case of TVC, the knife and cutting board surfaces show the greatest variation depending on the method. Due to the difficulty of making an imprint with the Petrifilm test on the knife surface, it can be presumed that the swabbing method is better for sampling small equipment, and the result is more accurate. In the case of the cutting board, due to the porosity of the material, the direct agar imprint method from the surface proved to be more effective in sampling.

Analysis of the *S. aureus* surface contamination results identified two clusters for both culture methods (Figure 5, lower). The Petrifilm method classifies the results of *S. aureus* contamination of the refrigerator surface as a separate cluster, similar to the level of contamination of the cutting and working board. In the case of detection of *S. aureus* using the reference method, comparable results of contamination of the tested surfaces were obtained. Only the working board showed lower contamination compared to the Petrifilm.

No significant correlations of *p* < 0.05 were found between the methods used to evaluate the contamination condition of *S. aureus*, whereas for TVC, there was a positive significant correlation between the results from the culture methods: reference and Petrifilm on the knife surface (r^2^ = 0.94) and between the reference method and adenosine 5-triphosphate values on the serving board (r^2^ = 0.72) (Figure 6). The results of visual hygiene assessments that were conducted during sampling for analysis in food truck outlets were also analysed. No correlation was found between the results obtained from the visual assessment and the results obtained from the other methods used in the study.

## 4. Discussion

### 4.1. Hygienic Status of Working Surfaces in Food Trucks

The hygienic statuses of the evaluated work surfaces in the food trucks varied (*p* < 0.05). The majority of the food trucks fulfilled the surface hygiene requirements; however, in seven of the food trucks, the cutting, serving, and working board surfaces were found to exceed the TVC of log 3 CFU/100 cm^2^, indicating high contamination of these surfaces. *Salmonella* spp. and *L. monocytogenes* were not detected in any of the food trucks assessed by the reference method (*n* = 20) despite the fact that chicken and eggs were used to prepare dishes in the two FT. However, the presence of *Listeria* spp. on the tested surfaces was detected by the alternative Petrifilm method in two food trucks. This does not indicate the presence of the pathogen but is rather an indicator of the existence of positive conditions for its growth. The reference plate method does not provide such information for 30 h (required incubation); therefore, the Petrifilm EL tests can be used to evaluate hygienic conditions in the establishment for the possible growth of pathogenic *L. monocytogenes*.

The risk to consumer health is also evidenced by the detection of *Enterobacteriaceae* and *E. coli* bacteria using culture methods on surfaces in the assessed food trucks. However, only one food truck (F10) showed levels of these bacteria above acceptable levels on all surfaces tested (refrigeration, cutting board, knife, serving board, and working board). High levels of contamination were also visible to the naked eye.

Other authors [13,14,66,67,68] have also reported problems with ensuring proper hygiene in catering establishments. According to them, poor food quality and inadequate hygiene conditions result in contamination with coliform bacteria, especially *E. coli* [14], which are hygiene indicators of faecal contamination in water and other production related environments [54,55,56,57]. The presence of *E. coli*, *S. aureus*, *Pseudomonas aeruginosa*, *Klebesiella*, and coliforms in the finished products shown in these studies is evidence of the poor hygiene and unsanitary practices used in the preparation and packaging of these street foods, as well as hand contamination among employees. These bacteria are indicators of a dirty environment, unhygienic pre- and post-production procedures, and poor water quality. Pathogenic bacteria are also carried by vegetables and street food, which infect workers, food handlers, and consumers in the industry. Previous studies [72,85,86,87] have also reported that any value greater than 1.0 log10 CFU/cm^2^ for total coliforms is not suitable for food preparation.

Contaminated food has a direct effect on human health, but contaminated surfaces (plates, mugs, cutting boards, working tables, and serving tables) are more critical. This is because contaminated surfaces can be one of the factors of food spoilage when RTE (ready-to-eat) food is in direct contact with these surfaces. These surfaces can become re-contaminated after routine cleaning procedures, and, in the case of RTE foods, they will no longer be cooked before being served to consumers. Consequently, equipment, utensils, and areas where food is processed or prepared require attention during cleaning or hygiene tasks so as not to achieve only apparent surface cleaning, which has been found to be the case in small food production facilities, such as food trucks [88]. According to Cooper et al. [86], the reason for inefficient cleaning and consequently higher ATP and TVC levels after cleaning processes may be the spread of microorganisms over the cleaned surface, especially when cleaned with reusable wipes [89,90,91].

In food service establishments, work areas, cutting boards, sinks, and kitchen taps are identified as key surfaces that can cause cross-contamination of food, particularly if these surfaces are contaminated by mesophilic aerobic bacteria and *Enterobacteriaceae* [71]. Many authors [72,73,74,92] have also identified significant numbers of TVCs and coliform bacteria taken from cutting boards, knives and spoons, slicers, tabletops, and tables in catering establishments which did not meet the standard for clean surfaces. These contaminations were caused by poor cleaning standards for these surfaces. Microbiological cleanliness for cutting boards depended on the length of time the boards had been in use; only new boards had high cleanliness levels. Boards that have been in use for a long time may have a damaged surface, which will be microbiologically contaminated despite properly conducted cleaning practices. Cutting boards have more irregular surfaces, so proper cleaning and disinfection are more difficult and favour the survival of biofilm-forming microorganisms [93,94]. The relevance of performing the washing operation with care is highlighted by Lee et al. [95]. They demonstrated the effectiveness of hand-washing knives inoculated with *Escherichia* and *Listeria innocua*, thereby obtaining a significant reduction in contamination levels, even at a low temperature and with a low concentration of disinfectant.

In the present study, surface contamination was detected above the acceptable level on refrigeration surfaces in only one case. Similar results are indicated by Czarniecka-Skubina [74], who found satisfactory microbiological quality of samples taken from refrigeration surfaces in 11 stationary catering establishments, finding only one species of coliform bacteria, namely *E. coli*. Contamination in refrigerators is significantly influenced by the type of product stored. The highest values of microorganisms were detected for the storage of raw meat and chicken meat, and the lowest were detected for vegetables and cooked products [69].

In conclusion, there is a strong link between contaminated surfaces in catering establishments and food safety risks.

### 4.2. Comparison of Control Methods in Food Trucks

Among the important factors for ensuring the safety and hygiene of street food, the knowledge and attitudes of street food vendors and their hygiene practices are crucial. For this reason, finding good control measures for this type of catering activity seems important.

The specifics of catering production make it difficult or even impossible to apply the same control methods in catering establishments as those applied in food industry facilities. In order to make control effective, a systematic approach and the prevention of hazards are essential. As other authors [96,97] point out, even very detailed controls of sanitary inspections are only a fraction of the operations carried out in catering establishments and will not prevent the risk of foodborne diseases. Routine inspections are difficult to conduct in food trucks because of the constant movement of facilities and the frequent change in the range of activities. Traditional microbiological methods for the detection and quantification of microorganisms on surfaces and equipment, which require culture and incubation, are not a good option in this case, as they are time-consuming and do not provide an immediate evaluation of the current state of hygiene in an establishment. In addition, food consumption takes place immediately after production, and it is pointless to obtain a result after this time. In the case of food trucks, preventive measures seem to be a more appropriate solution.

In the evaluated food trucks, a visual assessment was also carried out but was performed before the other methods, as it is inappropriate to use other methods of appraisal when surfaces are visibly dirty. Visual assessment obviously does not replace microbiological analyses. The results of the present study indicate that the visual assessment of the analysed surfaces was not correlated with the results of the microbiological analyses that were carried out using different methods. Visually clean surfaces may still have food residues or microorganisms, which result in food contamination. According to Tebbutt et al. [88], visual assessment underestimates actual surface contamination. When assessing the microbiologically clean surfaces of cutting boards, refrigerator door handles, and microwave oven control buttons, these authors found that they did not meet the conditions of hygienic cleanliness. A periodic visual inspection focusing on hygienic practices and microbiological supervision of surfaces that are at a high risk of cross-contamination provides valuable information for improving the knowledge, attitudes, and practices of food handlers towards food for better food safety [98].

Among the commercially available methods, there are rapid methods for detecting microbial or organic contamination on surfaces that results from improper hygiene processes. These include contact methods, bioluminescent methods, and modifications of plate methods, all of which were used in this study. According to some authors, easy-to-use microbial kits are practical, and the self-check approach in hygiene should be made mandatory or an alternative method for the operator [99].

One study [100] used environmental monitoring controls to look for potential correlations between microbiological indicators and food hygiene and sanitation conditions. It is impossible to completely eliminate pathogenic microorganisms from food production areas, but their growth, spread, and survival can be influenced by regular, thorough cleaning and disinfection of food contact surfaces as well as by monitoring their effectiveness. Surfaces can play a critical role in the development of food poisoning because of the potential for pathogens to grow on them. Then, these surfaces are handled by staff or consumers, and hands can be a medium for the transfer of bacteria and viruses to dishes and vice versa.

The reference method showed statistically significantly lower TVC and *S. aureus* contamination on the surfaces tested than the Petrifilm plate method. *S. aureus* was detected on the surfaces tested at levels not compatible with enterotoxin formation. Among the culture methods, the Petrifilm imprint plate method allowed for more effective recovery of *E. coli* and *Enterobacteriaceae* from the surfaces and better determination of their number compared to the swab method. Petrifilm EL tests provide a convenient tool for the mobile catering industry to assess the growth potential of pathogenic *L. monocytogenes*.

The evaluation of the hygiene status of street food with different methods did not demonstrate the alternate applicability of the culture methods, nor did the correlation between the subjective method and the measurement of adenosine 5-triphosphate.

Opinions vary among researchers on the suitability of alternative hygiene evaluation methods. Larson et al. [101] found no correlation between the bioluminescence method and microbiological reading values. According to these authors, the lower the amount of ATP, the lower the sensitivity of the method. Rosiak et al. [102] obtained a significant correlation between the results that were obtained by the ATP method and the results of TVC evaluation using Petrifilm^TM^ plates on the hand surface of food service personnel (r^2^ = 0.63) and work surfaces (r^2^ = 0.72). A significant correlation (r^2^ = 0.56) between TVC results obtained via the bioluminescence method and the reference method in the case of *E coli* bacteria and a weaker correlation (r^2^ = 0.30) on work surfaces between the bioluminescence method and the Petrifilm method in TVC studies were also obtained by Czarniecka-Skubina [74]. The high correlation of hygiene surface results that were obtained via the bioluminescence method in hospital kitchens and the reference method is also indicated by other authors [103]. Petrifilm™ tends to have a lower detection limit than other techniques used to evaluate surface contamination (i.e., swabbing methods) and is widely accepted and approved for microbiological analysis in the food and beverage industry [104].

The results obtained via the ATP method, which indicates the presence of food residues and microorganisms on surfaces, are obtained within 1 min, which is more efficient than surface monitoring using traditional microbiological methods. As highlighted by Aycicek et al. [103], the primary advantage of this method is that it can be used without a laboratory and without specialised personnel. However, it does not reflect quantitative microbiological detection on food contact surfaces. Traditional microbiological methods are cheaper but require more skill and time, and in catering, the result is needed immediately to take corrective action. Despite its many advantages, the ATP method does not quantify microorganisms on food contact surfaces and should be integrated with other techniques that help monitor surface hygiene [103].

Regardless of the level of strict inspection, the hygiene of food trucks in various countries is still unsatisfactory. Trends in a number of countries show that social media, smartphone applications, and online reviews of food trucks provide great opportunities to improve the hygiene practices of street food trucks. Some food standard agencies in several countries such as New Zealand, Australia, and the UK have a social media presence and recommend that customers download one of the food truck apps and look at customer reviews with respect to hygiene [105].

### 4.3. Limitations

One of limitations of this article is the sample size we considered in this study. We tried to obtain microbiological samples from more food trucks, but private owners refused to provide samples despite knowing that the samples would be obtained from clean surfaces. Food truck owners and employees feared fines.

## 5. Conclusions

Microbiological analyses that were conducted with two culture and alternative methods to assess the state of surface hygiene in food truck mobile catering establishments showed the presence of pathogenic bacteria *S. aureus* and *E. coli* and a risk of contamination by the pathogenic species *Listeria monocytogenes* and *Salmonella* spp., *Shigella* spp., *Klebsiella* spp., *Cronobacter* spp., *Serratia* spp., and *Citrobacter* spp. In the assessment of microbial contamination with bacteria, statistically higher results were obtained using the Petrifilm^TM^ PAC, STX tests. Furthermore, the recovery of *Enterobacteriaceae* and *E. coli* from the surface using Petrifilm^TM^ EB and EC was better than the swab method.

The results of the research indicate the need to constantly monitor the hygiene of surfaces in food trucks, given the fact that the use of mobile catering is becoming more and more popular due to convenience and price. In order to reduce the risk of foodborne infections caused by bacteria, it is important to introduce specific requirements for monitoring practices for the hygiene of food contact surfaces, in particular cutting boards and work surfaces. These studies did not support an alternative use of the highly convenient measurement of adenosine 5-triphosphate with culture methods and a method of visual assessment of hygiene status. The conducted analyses support the use of Petrifilm tests for routine surface monitoring in food trucks due to better recovery of bacteria from the surface, ease of performance, and interpretation of the results.

Another important cause of microbiological risk in mobile catering establishments is the lack of awareness of the employees. Efforts should focus on introducing mandatory, certified training for food truck personnel in the field of microbiological hazards, methods of hygienisation, and hygiene monitoring.

## Figures and Tables

**Figure 1 foods-12-00772-f001:**
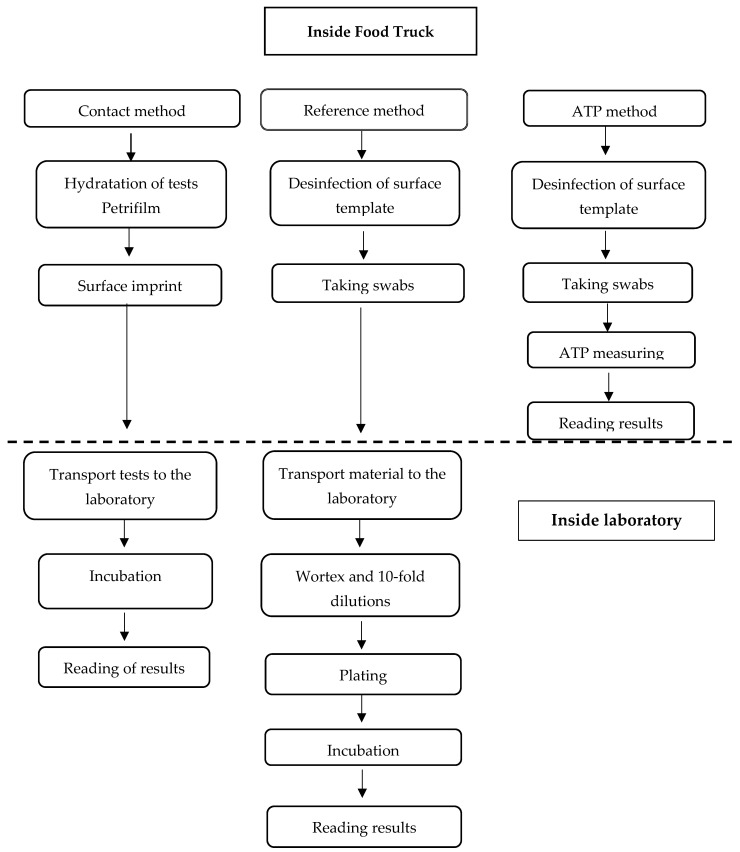
General scheme of analysis of hygiene state of food trucks.

**Figure 2 foods-12-00772-f002:**
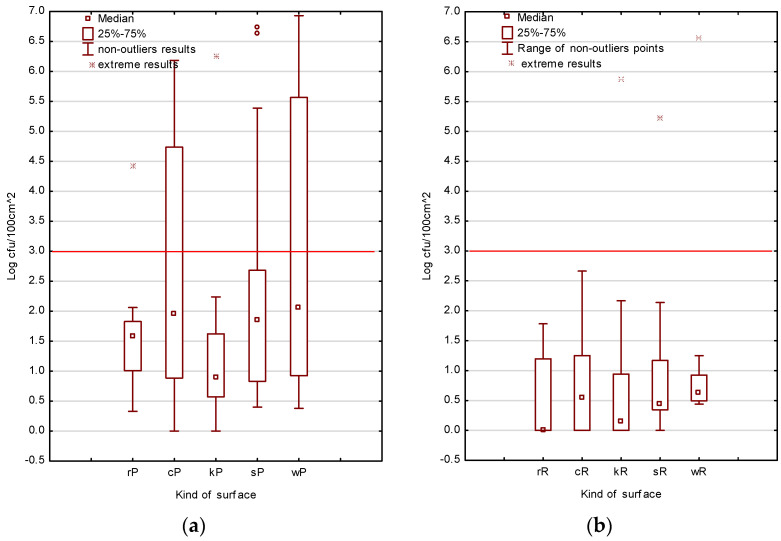
Mean contamination of TVC on analysed surfaces in food truck outlets (*n* = 20). (**a**) Petrifilm method (P); (**b**) Reference method (R). rP, rR—refrigerator; cP, cR—cutting board; kP, kR—knife; sP, sR—serving board; wP, wR—working board.

**Figure 3 foods-12-00772-f003:**
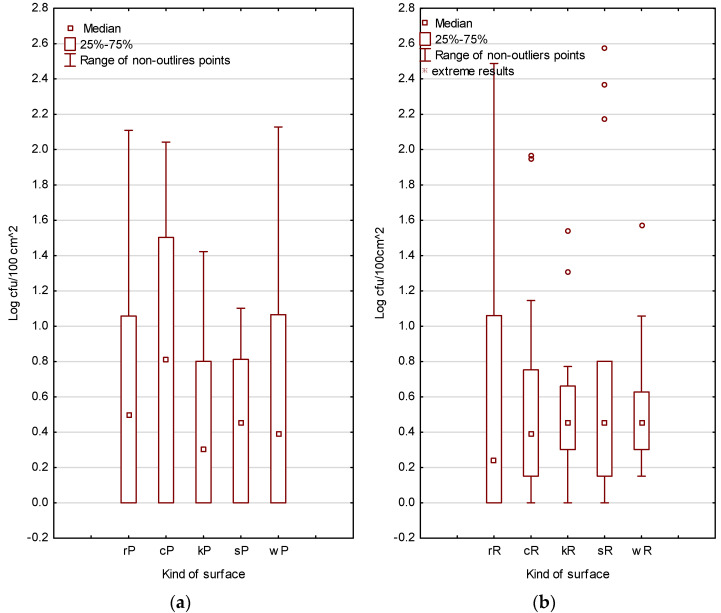
Mean contamination of *S. aureus* on analysed surfaces in food truck outlets (*n* = 20). (**a**) Petrifim method (P). (**b**) Reference method (R). rP, rR—refrigerator; cP, cR—cutting board; kP, kR—knife; sP, sR—serving board; wP, wR—working board.

**Figure 4 foods-12-00772-f004:**
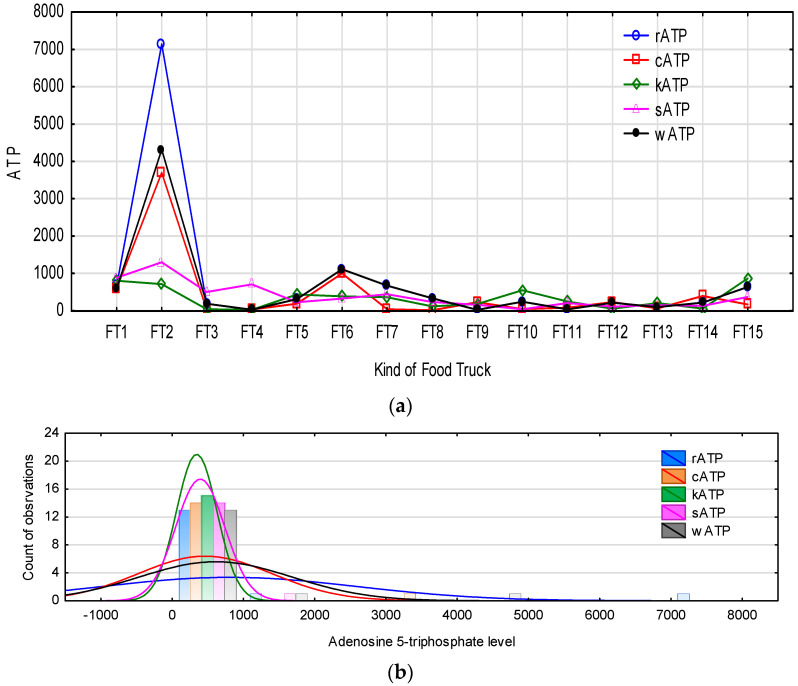
Level of adenosine 5-triphosphate on surfaces in food truck outlets, frequency and range of ATP results on analysed surfaces: r—refrigerator; c—cutting board; k—knife; s—serving board; w—working board. (**a**) Level of ATP on surfaces in food trucks outlet. (**b**) The frequency and range of ATP results on the analysed surfaces.

**Figure 5 foods-12-00772-f005:**
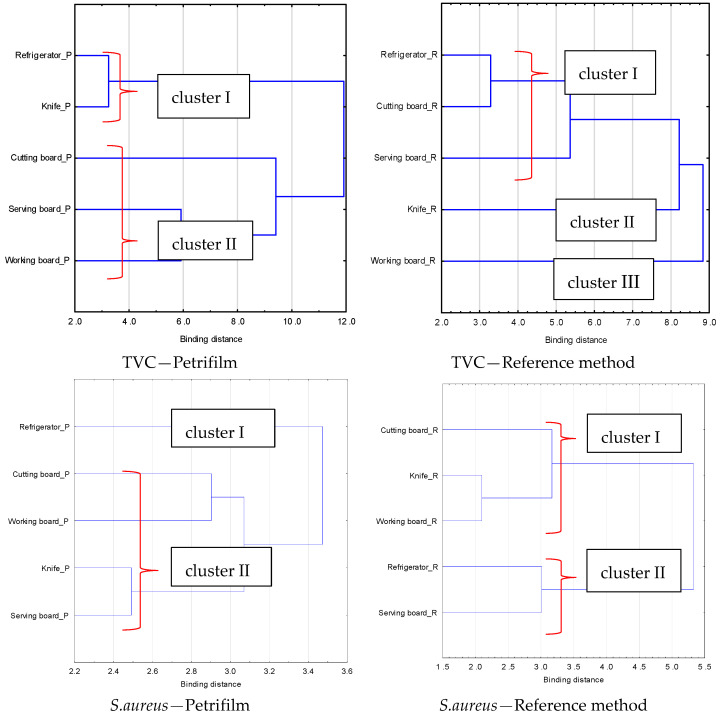
Cluster of surfaces depend on contamination level in analysed food truck outlets.

**Figure 6 foods-12-00772-f006:**
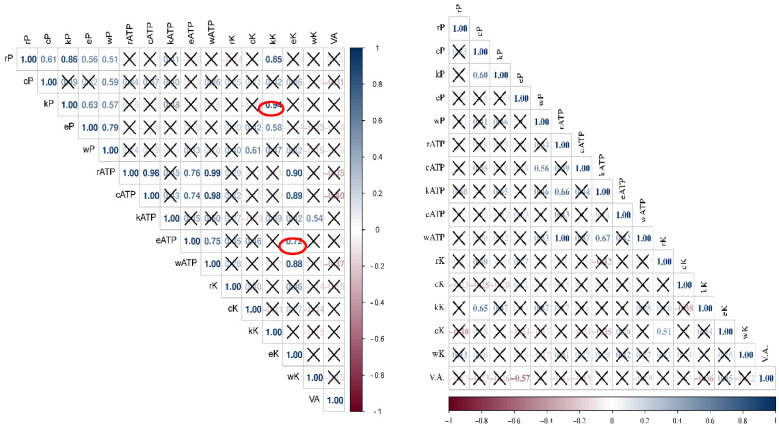
Heat map representing the correlation coefficient between the utilized methods for hygiene assessment in food trucks outlets. Upper triangle—TVC; lower triangle—*S. aureus*; P—Petrifilm method; ATP—adenosine 5-triphosphate; K—reference method; VA—visual assessment method; r—refrigerator; c—cutting board; k—knife; e—serving board; w—working board. X—label represents insignificant *p*-value.

**Table 1 foods-12-00772-t001:** Kind and characteristics of reference methods.

Kind of Microorganisms	Medium/Producer	ISO Standard	Typical Growth of Colonies
Total Viable Count (TVC)	NUTRIENT AGAR/ Neogen Co., Heywood, UK	[76]	All colony beside of shape, colour, size
*Staphylococcus aureus*	BARID PARKER/ BioRad, Watford, UK	[77]	Black-grey with transparent halo
*Enterobacteriaceae*	VRBG/ Oxoid Ltd., Hampshire, UK	[78]	Pink-violet
*Escherichia coli*	TBX/Oxoid Ltd., Hampshire, UK	[79]	Blue-green
*Listeria monocytogenes*	AL OA, PALCAM/ Neogen Co., Heywood, UK	[80]	Blue-green with cloudy halo; olive-grey with black centre
*Salmonella* spp.	BGA, XLD/Neogen Co., Heywood, UK	[81]	Black, red, pinkish, or white with red halo

**Table 2 foods-12-00772-t002:** Kinds and characteristics of contact plate method 3M™Petrifilm™.

Type of Petrifilm (Surface cm^2^)	Kind of Microorganisms	Medium Composition	Incubation Conditions	Typical Colonies
PAC (20)	TVC	Nutrient agents, tetrazol indicator	30 °C, 72 h	Pink-red
STX (30)	*Staphylococcus* *aureus*	Baird-Parker medium	35–37 °C, 24 h	Intensive red-purple
EL (40)	*Environmental Listeria* spp.	Selective and nutrient agents, chromogenic indicator	37 °C 26–30 h	Grey-purple
EB (20)	*Enterobacteriaceae*	VRBG medium	37 °C 24 h	Red with gas bubbles or with yellow acid zone or both
EC (20)	*Escherichia coli* and *coliforms*	VRBL medium	35–44 °C, 24–48 h	Blue or red-blue with gas bubble

**Table 3 foods-12-00772-t003:** The characteristics of the research material.

No. FT	Kind of Offer	Surface Details	Score of Visual Assessments *
FT1	Pizza	The surfaces in the food truck were new, composed of stainless steel and plastic; a round knife designed for cutting pizza was used for evaluation.	5
FT7	Fried chicken	All surfaces were clean and well maintained. The cutting board was composed of hard plastic.	5
FT12	Casseroles	All surfaces were of very good quality and cleanliness; cleaning took place just before testing.	5
FT4	Thai meals	In food truck, the equipment and surfaces were new, but not properly maintained; there were visible traces of dirt from the previous day’s work.	4
FT5	Greek meals	The food truck was not new; there were signs of contamination; however, all surfaces and equipment were in good condition—no visible signs of dirt, etc.	4
FT8	Burgers	In the food truck, most of the surfaces were clean and well maintained. The food truck was freshly renovated, but the cutting knives were dirty (there were visible traces of their previous use).	4
FT9	Burgers	Equipment and surfaces were maintained in good condition and clean. The cutting board was composed of the wrong material (wooden).	4
FT11	Casseroles	The working surfaces, as well as the production equipment, were kept in good condition. The consumer areas (3 tables) were not clean; they had not been cleaned the day before.	4
FT16	Burgers	All surfaces were of average quality; it was evident that they had been intensively used, but there was no visual dirt.	4
FT17	Burgers	All surfaces were of average quality; it was evident that they had been intensively used, but there was no visual dirt.	4
FT18	Burgers	All surfaces were of average quality; it was evident that they had been intensively used, but there was no visual dirt.	4
FT19	Burgers	All surfaces were of average quality; it was evident that they had been intensively used, but there was no visual dirt.	4
FT20	Burgers	All surfaces were of average quality; it was evident that they had been intensively used, but there was no visual dirt.	4
FT10	French fries	The surfaces from which the swab was taken were not of poor quality, whereas the other surfaces were dirty. It was noticed that the “old” frying oil was to be used (there were visible remnants of previous frying, the colour of oil was dark orange).	3
FT13	Ice cream	There were visible marks of raw material remaining after previous work the day before. The condition of the surfaces was average; defects in surface quality and cleanliness were visible.	3
FT15	Burgers	In the food truck, most of the surface was kept in good condition, except for the fridge; dirt and raw material residues were visible.	3
FT2	Israeli meals	The surfaces, which were composed of wood and plastic, had traces of many years of use.	2
FT3	Burgers	Inside the food truck, it was clear that there had been many years of operation without renovation; the walls were covered with dried fat. Small equipment, i.e., a cutting board and knives, were newly purchased.	2
FT14	Burgers	The equipment used in the street food outlets was not of good quality; the electric grill showed traces of burnt fat, the small appliances were of better quality (there were no visible signs of dirt), and the cutting board was composed of bamboo and was very wet	2
FT6	Ramen	The surfaces were new but not clean; traces of use were visible. The cutting board was composed of compressed bamboo; the worktops were composed of stainless steel.	1

FT—food truck, * subjective, visual assessment of hygiene on a scale 1–5 (1—poor hygiene state; 2—unsatisfactory hygiene state; 3—low satisfactory hygiene state; 4—good hygiene state; 5—very good hygiene state).

**Table 4 foods-12-00772-t004:** Number of surfaces contaminated by faecal bacteria *E. coli* and *Enterobacteriaceae* enumerated in food truck outlets by growth culture methods.

Method/Analysis	Food Truck (FT) *										Sum
	1	2	3	4	5	6	9	10	11	12	
Petrifilm/*Enterobacteriaceae*											
>1 log CFU	2		2	2	2	2	1	-	2	2	15
1–2 log CFU	2	3	-	-	-	-	-	-	-	-	5
<2 log	-	-	-	-	-	-	-	5	-	-	5
Petrifilm/*E. coli*											
>1 log CFU	2	1	-	-	-	-	-	-	-	-	3
1–2 log CFU	2	-	-	-	-	-	-	-	-	-	2
Reference/*Enterobacteriaceae*											
>1 log CFU	2	3	1	1	1	-	-	-	-	-	8
1–2 log CFU	1	-	-	-	-	-	-	-	-	-	1
Reference/*E. coli*											
>1 log CFU	-	-	-	-	-	-	-	-	-	-	0
1–2 log CFU	2	-	-	--	-	-	-	-	-	-	2
Sum of surfaces *Enterobacteriaceae* contaminated P/R Sum of surfaces *E. coli* contaminated P/R	4/3 4/2	3/3 1/0	2/1 0/0	2/1 0/0	2/1 0/0	2/0 0/0	1/0 0/0	5/0 0/0	2/0 0/0	2/0 0/0	

P—Petrifilm; R—reference; “-“—not detected; * FT in which the presence of *E. coli* and *Enterobacteriaceae* was found.

## Data Availability

All related data and methods are presented in this paper. Additional inquiries should be addressed to the corresponding author.

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
