# Peer review of "Surface Hygiene Evaluation Method in Food Trucks as an Important Factor in the Assessment of Microbiological Risks in Mobile Gastronomy"

_foods, 2023, doi:10.3390/foods12040772_

Round 1

Reviewer 1 Report

My comments to the authors can be found in the attached document.

Author Response

Authors’ Response to the Reviewers’ Comments

Journal:                             Foods

Manuscript:                     foods-2192657

Title:                                  Surface hygiene evaluation method in food trucks as an important factor in the assessment of microbiology risks in mobile gastronomy

Authors:                            MichaÅ‚ Wiatrowski, Elżbieta Rosiak, Ewa Czarniecka-Skubina

Article type:                    research article

2 February 2023

Dear Ms. Danika Zhang Assistant Editor, Reviewers,

We are excited to have been given the opportunity to resubmit the revised version of our manuscript (ID: foods-2192657) entitled ‘Surface hygiene evaluation method in food trucks as an important factor in the assessment of microbiology risks in mobile gastronomy’ Previously: ‘Hygiene evaluation method in food trucks as an important factor in the assessment of microbiology risks in mobile gastronomy’.

We greatly appreciate the time and efforts taken by the Reviewers and the Editor to review our manuscript.

We have addressed the issues indicated by Reviewers, and believe that the revised version meets the  journal publication requirements  and can be published in Foods Journal.

We agree with all suggestions and we tried to address them accordingly.

Please find our responses to the Reviewers' comments attached. The manuscript has been corrected for language errors by proofreader Stephen Fox-Hulme from ProofReaders.pl. All changes in the manuscript are highlighted in red.

 Your sincerely,

Ewa Czarniecka-Skubina

Reviewer 2 Report

See the comments in the attached file, need to address point by point.

Author Response

(The authors gave the same response as above.)

Reviewer 3 Report

Dear Authors,

The original manuscript entitled “Hygiene evaluation method in food trucks as important factor in assessment of microbiology risks in mobile gastronomy” is suitably well structured, developed and written by Wiatrowski et al. in appropriate English with a clear structure. They evaluated the hygiene of surfaces in food trucks (20 food trucks) by using conventional, Petrifilm and bioluminescence methods. They observed significant results and the study was generally interesting. Some major points should be considered in this manuscript. This manuscript should be reconsidered for a second review.

-        There are several spelling, punctuation and English grammar errors throughout the manuscript.

-        Your results should be more included in the abstract section.

-        There are several keywords used in this study. Please decrease them to 5 or 6 keywords. Some of them can be merged together.

-        Introduction section is too long and this section should be shortened maximumly of 5 individual paragraphs.

-        Is there any limitation in your study? What is the advantage of the methods you used with molecular techniques such as PCR? Viable cells also can be detected by using RT-PCR and specific mRNA in the samples. These methods are also rapid and reliable. Why you did not use these methods in your study? Explain it in your discussion section. Explain your limitations. One of your limitations is the sample size you considered in your study. If you cannot explain this limitation this manuscript should be regarded as a communication.

-        You can have more statistical analysis. It is highly recommended to measure the significant correlation among the presence of different pathogens in samples. You should improve your data analysis and scientific hypothesis through these methods. 

Author Response

(The authors gave the same response as above.)

Round 2

Reviewer 2 Report

No comment

Reviewer 3 Report

Dear Authors,

Thank you for your reply. All revisions have been addressed successfully.